# Impact of Maternal and Offspring Dietary Zn Supplementation on Growth Performance and Antioxidant and Immune Function of Offspring Broilers

**DOI:** 10.3390/antiox11122456

**Published:** 2022-12-14

**Authors:** Yuanyuan Wang, Ling Zhang, Yibin Xu, Xiaoqing Ding, Yongxia Wang, Aikun Fu, Xiuan Zhan

**Affiliations:** 1College of Animal Sciences, Zhejiang University, Hangzhou 310058, China; 2College of Animal Sciences, Anhui Agricultural University, Hefei 230036, China; 3College of Animal Science and Technology, Zhejiang A & F University, Linan 311300, China

**Keywords:** maternal zinc, zinc glycinate, antioxidant, immune, broiler

## Abstract

The current study investigated the effects of the maternal Zn source in conjunction with their offspring’s dietary Zn supplementation on the growth performance, antioxidant status, Zn concentration, and immune function of the offspring. It also explored whether there is an interaction between maternal Zn and their offspring’s dietary Zn. One-day-old Lingnan Yellow-feathered broilers (n = 800) were completely randomized (n = 4) between two maternal dietary supplemental Zn sources [maternal Zn–Gly (oZn) vs. maternal ZnSO_4_ (iZn)] × two offspring dietary supplemental Zn doses [Zn-unsupplemented control diet (CON), the control diet + 80 mg of Zn/kg of diet as ZnSO_4_]. oZn increased progeny ADG and decreased offspring mortality across all periods, especially during the late periods (*p* < 0.05). The offspring diet supplemented with Zn significantly improved ADG and decreased offspring mortality over the whole period compared with the CON group (*p* < 0.05). There were significant interactions between the maternal Zn source and offspring dietary Zn with regards to progeny mortality during the late phase and across all phases as a whole (*p* < 0.05). Compared with the iZn group, the oZn treatment significantly increased progeny liver and serum Zn concentrations; antioxidant capacity in the liver, muscle, and serum; and the IgM concentration in serum; while also decreasing progeny serum IL-1 and TNF-α cytokine secretions (*p* < 0.05). Similar results were observed when the offspring diet was supplemented with Zn compared with the CON group; moreover, adding Zn to the offspring diet alleviated progeny stress by decreasing corticosterone levels in the serum when compared to the CON group (*p* < 0.05). In conclusion, maternal Zn–Gly supplementation increased progeny performance and decreased progeny mortality and stress by increasing progeny Zn concentration, antioxidant capacity, and immune function compared with the same Zn levels from ZnSO_4_. Simultaneously, Zn supplementation in the progeny’s diet is necessary for the growth of broilers.

## 1. Introduction

Zinc (Zn) is an essential mineral that is usually added to the diets of the breeder to obtain the best reproductive performance. It ensures the development of bones and feathers in birds, as well as fertility and scavenging of oxygen free radicals in animals [1,2]. Additionally, Zn is one of the most important active elements for transforming protein, carbohydrate, nucleic acid, and activating enzyme, which is essential for the proper operation of the immune system [3,4,5].

Maternal effects are one of the most important epigenetic inheritance phenomena, and can directly influence the offspring’s growth, development, nutrient metabolism, and physiological characteristics [6]. Previous studies reported that maternal stress not only influences the birth weight and growth of offspring, but also changes the metabolic parameters and hormone secretion of mammals [7]. Zhu et al. [8] indicated that maternal Zn supplementation, regardless of the source of Zn, can reduce the negative impact of maternal heat stress on the progeny’s growth performance, and maternal organic Zn supplementation can improve the meat quality of offspring. For poultry, hatching eggs are an important pathway for transgenerational nutritional regulation. Maternal dietary Zn supplementation increased the Zn content in egg yolk [9], alleviated progeny intestinal inflammation, and improved the skeletal muscle development of offspring [10,11]. Therefore, as one kind of important feed additive, Zn is widely used in poultry feed. Inorganic Zn (including sulfate, oxide salts, etc.,) is one of several important main forms of Zn supplementation in poultry diets. However, antagonism can occur between inorganic Zn and other inorganic trace minerals [12]. The lower bioavailability of inorganic Zn results in its excessive addition, which causes waste and even environmental pollution [13]; however, organic Zn has a higher bioavailability than inorganic Zn [14,15].

Glycine (Gly) chelate is composed of one to three Gly particles combined with Zn, which determines its strong stability and chemical and physical homogeneity. Therefore, Gly has a better stability constant for Zn, which may facilitate easier uptake by the Zn–Gly complex [16]. Our previous studies found that the optimal dosage for broiler breeders was 80 mg Zn/kg of Zn–Gly [17,18], however, little research has been done about maternal Zn in relation to their offspring’s dietary Zn supplementation in broilers. We hypothesized that maternal organic Zn–Gly might have a better effect than inorganic zinc sulfate (ZnSO_4_) on their offspring’s growth performance, antioxidant status, immune function, etc., and expect to observe an interaction between maternal Zn and their offspring’s dietary Zn. Therefore, in this study, the main purpose was to explore the effects of maternal Zn in conjunction with their offspring’s dietary Zn supplementation on growth performance, antioxidant status, Zn concentration, and immune function of offspring from female breeders subjected to 80 mg of Zn/kg of diet from ZnSO_4_ and Zn–Gly, respectively. We have also explored whether there is an interaction between maternal Zn and their offspring’s dietary Zn.

## 2. Materials and Methods

### 2.1. Experimental Design, Birds, and Diets

This study was approved by the Animal Care and Use Committee of Zhejiang University. Experimental design and treatments for the maternal broiler breeder phase were discussed in detail in our previous study [18]. A completely randomized design was performed with two maternal dietary supplemental Zn sources [maternal Zn–Gly (oZn) vs. maternal ZnSO_4_ (iZn)] × two offspring dietary supplemental Zn doses [Zn-unsupplemented control diet (CON), the control diet + 80 mg of Zn/kg of diet from ZnSO_4_ (80 mg zinc/kg was derived from crystal standard zinc, provided by PANCOSMA, Switzerland). Firstly, a total of 200 39-week-old Lingnan Yellow broiler breeders were randomly divided into two groups—the oZn group and iZn group—with each group containing 5 replicates with 20 birds each. All broiler breeders from the two groups were fed a corn–soybean meal basal diet with 24 mg Zn/kg for four weeks pre-test to consume excess Zn in the body; in the formal feeding period of eight weeks, the iZn group and oZn group were supplemented with 80 mg Zn/kg from ZnSO_4_ and 80 mg Zn/kg from the Zn–Gly-based basal diet, respectively. All eggs from the last seven days were collected from two breeder groups (oZn group and iZn group) for offspring incubation. Then, 400 healthy offspring chicks from each group were selected and divided into two groups with two offspring dietary supplemental Zn doses (0 or 80 mg Zn/kg). Figure 1 shows the entirety of the experimental design. Table 1 displays the composition and nutrient content of the basal diet for the broiler breeders. The control group received the basal diet without Zn supplementation (negative control), and (iZn-iZn) and Zn–Gly (oZn-oZn) received a basal diet supplemented with 80 mg of Zn/kg of diet from ZnSO_4_ and Zn–Gly, respectively. Then, all hatched chicks were reared for the 60 day offspring broiler phase. Table 2 shows the composition and nutrient level of the basal diet for the offspring broiler. On days 21 and 60, three male broilers from each replicate were randomly selected and slaughtered. Serum samples from the blood were isolated by centrifuging at 3000× *g* for 10 min and stored at −80 °C until analysis. Liver and left breast muscle samples were immediately stored at −80 °C after slaughter for further analysis.

### 2.2. Measurement of Growth Performance and Sample Collection

On days 1, 21, and 60 of the experiment, the birds and their diet were weighed per the whole replicate. The variables of the growth performance (final body weight (BW)), average daily feed intake (ADFI), average daily gain (ADG), and feed:gain ratio (F:G) were measured.

### 2.3. Zn Concentration Determination

As described in our previous study [17], inductively coupled plasma atomic emission spectrometry (Iris Intrepid; Thermo Elemental, Waltham, MA, USA) was used to determine the concentrations of Zn in serum, liver, and breast muscle after microwave wet digestion in nitric acid (MARS 5; CEM Corp., Matthews, NC, USA) and dilution with distilled water. Calibration of Zn content determination was performed using a series of mixtures containing Zn standard solutions of graded concentrations [19].

### 2.4. Antioxidant Status Analysis

After combining the liver and muscle of each replicating broiler into a separate sample, the sample was weighed and then homogenized into nine volumes of sodium chloride solution (0.86%) (4000 rpm, 4 °C, 15 min) with Ultra Turrax homogenizer (Tekmar Co., Cincinnati, OH, USA). The homogenates were then centrifuged at 3500 rpm at 4 °C for 15 min and the supernatant was collected for further analysis [18].

As demonstrated in our previous study, the total superoxide dismutase (T-SOD) activities, copper zinc superoxide dismutase (Cu-Zn SOD) activities, total antioxidant capacity (T-AOC), and malondialdehyde (MDA) concentrations were evaluated using a commercial analytical kit (Nanjing Jiancheng Bioengineering Institute, Nanjing, China). The results were standardized with the total protein for internal comparison. The concentrations of liver metallothionein (MT) was determined by the commercial ELISA kit purchased from Shanghai Elisa Technology Co., Ltd. All procedures were carried out according to the instructions of the assay kits [18].

### 2.5. Chemical and Cytokine Secretions Analyses

Immunoglobulins (IgA, IgG and IgM) and pro-inflammatory cytokines (IL-1, IL-2, and TNF-α) were determined using the commercial ELISA kit purchased from Shanghai Elisa Technology Co., Ltd. All procedures were carried out according to the instructions of the assay kits.

### 2.6. Statistical Analysis

All data were analyzed by two-way ANOVA using the general linear model procedure of SAS 9.2 (SAS Institute, 2010, Cary, NC, USA). This model combines the main effects of maternal dietary Zn, progeny dietary Zn, and their interaction. The treatment comparisons for significant differences were tested by the LSD method. Each replicate served as the experimental unit for all statistical analyses. Significant differences were set at *p* ≤ 0.05.

## 3. Results

### 3.1. Growth Performance

During days 1–60, oZn increased the progeny ADG and decreased the mortality (*p* < 0.05), but no significant differences were observed for the ADFI and F/G parameters. The offspring diet supplemented with Zn improved the ADG by 2.55% and significantly decreased mortality compared with the CON group (*p* < 0.05). Significant interactions between maternal Zn and offspring’s dietary Zn supplementation were observed in progeny mortality during the entire phase (Table 3).

### 3.2. Zn Concentrations

The effects of maternal Zn in conjunction with their offspring’s dietary Zn supplementation on the serum, liver, and muscle Zn concentrations of offspring broilers are shown in Table 4. Maternal dietary Zn as Zn–Gly rather than ZnSO_4_ supplementation resulted in higher Zn concentrations in the liver of the offspring at 21 days of age and increased progeny Zn concentrations in serum, liver, and muscle at 60 days of age (*p* < 0.05). Additionally, the broiler offspring’s dietary 80 mg/kg of Zn supplementation increased the Zn concentrations in the serum, liver, and muscle during the entire phase compared to the CON group (*p* < 0.05). Significant interactions (*p* = 0.031) between maternal Zn and offspring’s dietary Zn supplementation were observed in the progeny’s muscle at 60 days of age.

### 3.3. Antioxidant Status in Serum

The effects of maternal Zn in conjunction with their offspring’s dietary Zn supplementation on the serum antioxidant status of offspring broilers are shown in Table 5. Maternal dietary oZn supplementation increased the activities of T-AOC (by 5.66%), T-SOD (by 8.58%), and CuZn-SOD (by 6.24) compared to iZn at 21 days of age, and similar results were also observed in the broiler’s offspring dietary 80 mg Zn/kg supplementation compared with the CON group (*p* < 0.05). At 60 days of age, T-AOC activity increased by 7.99% in the case of oZn compared to iZn; and the activity of T-AOC (by 21.49%) and CuZn-SOD (by 6.43%) significantly increased when the broiler’s offspring dietary 80 mg Zn/kg supplementation (*p* < 0.05) compared with the CON group. The MDA content decreased in offspring broilers at 21 and 60 days of age compared to maternal ZnSO_4_ (*p* < 0.05). Significant interactions between maternal Zn and offspring dietary Zn supplementation were observed in progeny MDA content at 21 days of age and activity of T-SOD at 60 days of age (*p* < 0.05).

### 3.4. Antioxidant Status in Liver

The effects of maternal Zn in conjunction with their offspring’s dietary Zn supplementation on the liver antioxidant status of offspring broilers are shown in Table 6. Maternal dietary Zn–Gly supplementation increased the T-SOD activity and MT content in offspring broilers at 21 and 60 days of age and CuZn-SOD activity in offspring broilers at 21 days of age compared with that of maternal ZnSO_4_ (*p* < 0.05), whereas maternal Zn supplementation did not affect their offspring regarding the activity of T-AOC in the liver of the progeny (*p* > 0.05). Additionally, compared with the CON group, offspring broilers dietary 80 mg/kg of Zn supplementation increased progeny T-AOC, T-SOD, and CuZn-SOD activities at 21 days of age, and decreased progeny MDA content and increased MT content at 21 and 60 days of age (*p* < 0.05). Significant interactions (*p* < 0.001) between maternal Zn and offspring dietary Zn supplementation were observed in the progeny’s MDA content at 21 days of age and the progeny’s T-AOC, T-SOD, and CuZn-SOD activities at 60 days of age (*p* < 0.05).

### 3.5. Antioxidant Status in Muscle

The effects of maternal Zn in conjunction with their offspring’s dietary Zn supplementation on the muscle antioxidant status of broiler offspring are shown in Table 7. Maternal dietary Zn as Zn–Gly rather than ZnSO_4_ supplementation increased T-AOC activity and decreased MDA content at both 21 and 60 days of age (*p* < 0.05). Furthermore, their offspring broiler’s dietary 80 mg/kg of Zn increased the activities of T-SOD and CuZn-SOD at 21 days of age, and increased the T-AOC activity and decreased MDA content at both 21 and 60 days of age (*p* < 0.05). Meanwhile, no significant interactions between maternal Zn and offspring dietary Zn supplementation were observed in the antioxidant status of the progeny’s muscle (*p* > 0.05).

### 3.6. Cytokine Secretion

The effects of maternal Zn in conjunction with their offspring’s dietary Zn supplementation on the cytokine secretion of offspring broilers are shown in Table 8. Maternal dietary Zn as Zn–Gly rather than ZnSO_4_ supplementation had no significant effects on progeny IL-1, IL-2, and TNF-α concentrations at 21 days of age while significantly decreasing the IL-1 and TNF-α concentrations of progeny at 60 days of age (*p* < 0.05). Similar results were observed for offspring broilers dietary 80 mg/kg of Zn. No significant interactions between maternal Zn and dietary Zn supplementation were observed in progeny cytokine secretions at 21 days of age (*p* > 0.05), whereas significant interactions were observed in progeny IL-1 (*p* = 0.012) and TNF-α (*p* = 0.030) concentrations at 60 days of age.

### 3.7. Immunoglobulin Content

The effects of maternal Zn in conjunction with their offspring’s dietary Zn supplementation on the immunoglobulin content of offspring broilers are shown in Table 9. Maternal dietary Zn–Gly rather than ZnSO_4_ supplementation increased progeny IgM concentrations at both 21 and 60 days of age (*p* < 0.05). Additionally, offspring dietary supplementation of 80 mg/kg of Zn increased the IgA content of progeny at 21 days and the IgG and IgM concentrations of progeny at 60 days of age (*p* < 0.05); moreover, no significant interactions between maternal Zn and offspring dietary Zn supplementation were observed in progeny immunoglobulin content (*p* > 0.05).

### 3.8. Stress Index

The effects of maternal Zn in combination with their offspring’s dietary Zn supplementation on the stress index of the broiler’s offspring are shown in Table 10. Maternal Zn–Gly rather than ZnSO_4_ supplementation had no significant effects on progeny CORT, HSP70, and CK content at 21 days of age; however, it significantly decreased the CORT content of progeny at 60 days of age. Additionally, their broiler’s offspring dietary supplementation of 80 mg/kg of Zn significantly decreased the CORT content at 21 days of age (*p* < 0.05), and Zn–Gly rather than ZnSO_4_ supplementation significantly increased the HSP70 content at 60 days of age (*p* < 0.05). Significant interactions between maternal Zn and offspring’s dietary Zn supplementation were observed in progeny CORT (*p* = 0.033) and HSP70 (*p* = 0.021) concentrations at 60 days of age.

## 4. Discussion

Maternal Zn, in combination with their offspring’s dietary Zn supplementation, had no significant effects on the progeny growth performance during the stage of days 1–21. Furthermore, supplementation of 80 mg Zn/kg from Zn–Gly increased progeny BW at 60 days as well as ADG during the stage of days 22–60, and had a lower mortality rate compared to the same Zn levels from ZnSO_4_, thereby indicating that maternal dietary organic Zn in conjunction with their offspring’s dietary Zn supplementation improved the growth performance of progeny in the final period. Consistent with our findings, several studies reported that dietary organic Zn supplementation increased growth rate, feed intake, and feed efficiency in broiler chicks [20,21,22,23] however, reports showed that low Zn diets will lead to loss of appetite, feed intake, and weight gain [24,25], resulting in decreased hatchability and increased embryonic mortality [26,27] and even further depressed the growth performance of progeny [8]. It is implied that in the current study, maternal dietary Zn supplementation may be transferred to progeny broilers during the stage of embryonic development and showed in the progeny growth performance of the final period.

Dietary Zn supplementation in broiler breeders markedly enhanced the Zn concentrations of eggs [9,28]. In our previous studies, the Zn concentrations in muscle and eggs [17] increased with dietary organic Zn supplementation compared with inorganic Zn and even increased in the livers of developing embryos and one-day-old chicks [18] moreover, we found that maternal 80 mg Zn/kg from Zn–Gly increased the Zn concentration in progeny serum, liver, and muscle compared to the same Zn levels from ZnSO_4_. This may be because in broilers, dietary organic Zn supplementation had a higher absorption rate and lower excretion rate than inorganic Zn [29], thereby resulting in higher Zn concentrations. Interestingly, the broiler offspring’s dietary Zn supplementation increased Zn concentrations in their liver at 21 days of age and remained stable over time. On the contrary, it returned to initial values or was even lower in serum and muscle at 60 days of age. This may be because broilers were at different growth stages and had different nutritional requirements. The increased deposition of Zn in egg yolk leads to higher Zn availability, which facilitates the growth of chick embryos [30]. Consistent with the results in our present study, Kwiecień et al. [1] indicated that organic Zn addition enhanced the Zn concentrations in the liver of broilers compared to inorganic Zn supplementation, implying that the results of increased egg, progeny tissue, and serum Zn concentrations were from the increased maternal Zn.

Given the better antioxidant properties of Zn, therefore, dietary Zn status is closely related to the antioxidant system [31] it influences antioxidant status by a different mechanism: First, protecting proteins and enzymes against free radical attack or oxidation, and second, preventing free radical formation produced by other metals, such as iron and copper [32]. T-AOC, T-SOD, and MDA are marks of oxidative stress as they participate in the antioxidant defense systems that result in potential damages induced by oxidative stress [33]. Zn is an essential component of CuZn-SOD, and its potential antioxidant effects could be associated with its role in the mechanism of the structural integrity of CuZn-SOD [34]. Additionally, its presence in the Zn metallothionein protects organs and tissues against immune-mediated attacks by ROS [35].

Zn deficiency might cause an increase in the production of free radicals [36], resulting in decreased SOD activity and increased MDA content [37]. However, Zn supplementation enhanced the activities of CuZn-SOD and GSH-Px and decreased the MDA levels in broilers’ livers [38]. Our previous experiment showed that maternal Zn supplementation significantly reduced the content of MDA and enhanced the activity of T-SOD, CuZn-SOD, and T-AOC in the livers of both chick embryos and one-day-old chicks [18]. Further experiments showed that maternal Zn, in conjunction with their offspring’s dietary Zn supplementation, markedly reduced the MDA concentrations and enhanced the activity of T-AOC and CuZn-SOD in progeny serum, liver, and muscle, and enhanced MT concentrations in the liver of progeny. In addition, maternal organic Zn had better effects on the antioxidant defense of progeny serum and tissue than inorganic Zn. It is implied that maternal Zn, in combination with their offspring’s dietary Zn supplementation, had better effects on antioxidant capacity; it reduces the oxidative stress of offspring broilers by increasing the activity of CuZn-SOD and inducing the production of MT, thereby improving the production performance of broilers.

Zn can interact with the immune systems’ components and is thought to be essential for immune response [39]. It binds to enzymes, proteins, and peptides with different binding affinities [40]. A previous study reported that the immune organs, including the thymus, spleen, and bursa of fabricius of three-week-old broilers, were significantly influenced by Zn–Gly levels in their diets [41]. Additionally, Zn supplementation is an adjunctive therapy for treating chronic and inflammatory diseases; modulating TNF-α, IL-6, and IL-10; and modifying the gene expression of MT in white blood cells. This highlights the importance of Zn in gene transcription and cell metabolism [42,43]. It can also prevent Salmonella-induced liver injury by reducing pro-inflammatory cytokine levels [44]. In the current study, the results indicated that maternal organic Zn and their offspring’s dietary Zn supplementation significantly reduced the concentrations of IL-1 and TNF-α, suggesting that the protective effects of Zn on the broiler may be through downregulating the inflammation of the host; moreover, the severity of the inflammatory response is generally inversely proportional to the serum Zn concentration [45]. Hence, the additional possible mediator of Zn, which alleviates the host’s inflammatory response, is caused by the enhanced antioxidant defense and higher Zn concentration.

In the present study, maternal organic Zn and their offspring’s dietary Zn supplementation had significantly higher immunoglobulin concentrations and complements. Consistent with our present results, ample Zn supplementation improves the ability to produce antibodies [46,47]. Fraker et al. [48] showed that supplementing Zn as ZnSO_4_ to a diet deficient in Zn was effective for total serum IgG concentration, and diets supplemented with Zn as Zn–Gly had higher levels of immunoglobulins (IgA, IgM, and IgG) in the serum of broilers [41], moreover, Wellinghausen et al. [49] indicated that Zn status in Zn methionine and Zn sulfate affected the total non-specific IgM and IgG concentrations in seven-day-old broilers.

Stressors (such as heat stress and oxidative stress) can increase plasma corticosterone concentration in different tissues of poultry with a concomitant increase in creatine kinase levels [50]. In the present study, offspring dietary Zn supplementation significantly decreased the corticosterone levels, indicating that Zn supplementation can reduce oxidative stress during broiler production, as Zn supplementation increased broiler tissue Zn content—thereby improving its antioxidant status and reducing oxidative stress. However, we also found that Zn supplementation increased the levels of HSP70, which may be because HSP70 is the most important family of heat shock proteins and can produce non-specific tolerance to various stressors [51]. The increased levels of HSP70 in vivo alleviates the stress of environmental stressors on chickens to a certain extent.

In the current study, as this experiment was a 2 × 2 factor experimental design, the offspring broilers were only supplemented with inorganic Zn with different doses (0 or 80 mg Zn/kg). In our future experiments, we will explore the effects of adding different forms of Zn sources (including inorganic Zn and organic Zn) to the offspring broilers.

## 5. Conclusions

In conclusion, maternal Zn–Gly supplementation increased progeny performance and decreased progeny mortality and stress by increasing the progeny Zn concentration, antioxidant capacity, and immune function at the same Zn levels as ZnSO_4_. At the same time, Zn supplementation in the progeny’s diet is necessary for the growth of broilers.

## Figures and Tables

**Figure 1 antioxidants-11-02456-f001:**
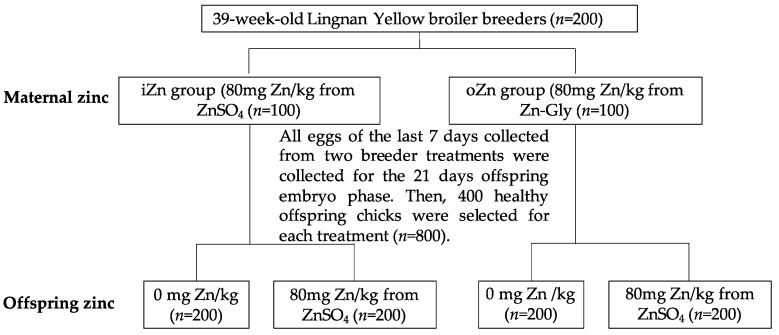
Experimental design of broilers.

**Table 1 antioxidants-11-02456-t001:** Composition and nutrient level of the basal diet for broiler breeders (%, unless otherwise stated).

Ingredients	
Maize	64.6
Soybean meal	25.0
CaHPO_4_	1.8
Limestone	7.0
Salt	0.3
DL-methionine	0.3
Premix ^1^	1.0
Nutrient Composition	
ME ^2^ (MJ/kg)	11.21
Crude protein	16.11
Calcium	3.04
Total phosphorus	0.63
Lysine	0.80
Met + Cys	0.82
Zn (mg/kg)	24

^1^ Supplied the following per kilogram of diet: iron, 80 mg; copper, 8 mg; selenium, 0.3 mg; manganese, 100 mg; iodine, 1.0 mg; vitamin A, 10,800 IU; vitamin D3, 2160 IU; vitamin E, 27 IU; vitamin K3, 1.4 mg; vitamin B1, 1.8 mg; vitamin B2, 8 mg; vitamin B6, 4.1 mg; vitamin B12, 4.1 mg; niacin, 25 mg; calcium pantothenate, 10 mg; folic acid, 0.55 mg; biotin, 0.15 mg. ^2^ ME was calculated from data provided by Feed Database in China.

**Table 2 antioxidants-11-02456-t002:** Composition and nutrient level of the basal diet for offspring broiler (%, unless otherwise stated).

Item	Starter Period 1 to 21d	Grower Period 22 to 42d	Finisher Period 43 to 60d
Maize	59.0	63.6	65.5
Soybean meal	36.0	31.0	29.0
Soybean oil	1.0	1.4	2.0
CaHPO_4_	1.5	1.2	1.0
Limestone	1.2	1.5	1.2
Salt	0.3	0.3	0.3
Premix ^1^	1.0	1.0	1.0
Total	100	100	100
Nutrient levels			
ME ^2^ (MJ/kg)	11.88	12.18	12.43
Crude protein	20.52	18.68	17.95
Lysine	1.21	1.05	1.02
Methionine	0.49	0.41	0.42
Met + Cys	0.81	0.73	0.71
Calcium	0.92	0.93	0.76
Total phosphorus	0.64	0.57	0.53
Zn (mg/kg)	28.0	26.6	26.0

^1^ Supplied the following per kilogram of diet: iron, 80 mg; copper, 8 mg; manganese, 80 mg; iodine, 0.35 mg; selenium, 0.15 mg; vitamin A, 5000 IU; vitamin D3, 1000 IU; vitamin E, 10 IU; vitamin K3, 0.5 mg; vitamin B1, 2 mg; vitamin B2, 3 mg; vitamin B6, 3 mg; vitamin B12, 0.01 mg; niacin, 25 mg; calcium pantothenate, 10 mg; folic acid, 0.55 mg; biotin, 0.15 mg. ^2^ ME was calculated from data provided by Feed Database in China.

**Table 3 antioxidants-11-02456-t003:** Effects of maternal Zn in conjunction with their offspring dietary Zn supplementation on growth performance of day 1–60 in offspring broilers (mean ± S.E.).

Maternal Zinc	Offspring Zinc (mg/kg)	ADG (g)	ADFI (g)	F/G	Mortality (%)
ZnSO_4_ ^1^	0	30.62 ± 0.22	76.15 ± 1.32	2.53 ± 0.04	10.00 ± 0.00 ^a^
	80	31.40 ± 0.27	73.22 ± 0.63	2.49 ± 0.01	2.67 ± 1.15 ^c^
Zn–Gly ^1^	0	31.33 ± 0.16	73.98 ± 3.71	2.49 ± 0.03	4.00 ± 0.00 ^b^
	80	32.11 ± 0.08	73.46 ± 1.72	2.49 ± 0.14	2.00 ± 0.00 ^c^
ZnSO_4_ ^2^		31.01 ± 0.48 ^b^	74.68 ± 1.86	2.51 ± 0.03	6.33 ± 4.08
Zn–Gly ^2^		31.72 ± 0.45 ^a^	73.72 ± 2.60	2.48 ± 0.09	3.00 ± 1.09
	0 ^3^	30.97 ± 0.43 ^b^	75.07 ± 2.76	2.51 ± 0.04	7.00 ± 3.29
	80 ^3^	31.76 ± 0.43 ^a^	73.34 ± 1.16	2.49 ± 0.09	2.33 ± 0.81
*p*-value	Maternal zinc	<0.001	0.464	0.600	<0.001
	Offspring zinc	<0.001	0.204	0.663	<0.001
	Maternal zinc × offspring zinc	0.988	0.364	0.649	<0.001

ADG, average daily gain. ADFI, average daily feed intake. F/G, feed/gain. Different letters (a–c) represent significant differences between the groups. ^1^ Each value represents the mean ± SE of 5 replicates (*n* = 200). ^2^ Each value represents the mean ± SE of 10 replicates (*n* = 400). ^3^ Each value represents the mean ± SE of 10 replicates (*n* = 400).

**Table 4 antioxidants-11-02456-t004:** Effects of maternal Zn in conjunction with their offspring dietary Zn supplementation on the Zn content in serum, liver, and muscle of offspring broilers (mean ± S.E.).

Maternal Zinc	Offspring Zinc (mg/kg)	21d			60d		
		Serum (μmol/L)	Liver (µg/g)	Muscle (µg/g)	Serum (μmol/L)	Liver (µg/g)	Muscle (µg/g)
ZnSO_4_ ^1^	0	40.29 ± 3.71	44.86 ± 3.79	38.77 ± 7.70	27.37 ± 1.07	52.36 ± 2.03	20.46 ± 3.28 ^b^
	80	44.84 ± 2.36	47.16 ± 2.87	49.84 ± 4.91	31.08 ± 0.99	57.25 ± 2.26	34.08 ± 4.89 ^a^
Zn–Gly ^1^	0 41.44 ± 2.95 45.92 ± 4.51 46.73 ± 2.22 29.90 ± 2.09 55.74 ± 2.59 32.30 ± 3.02 ^a^	41.44 ± 2.95	45.92 ± 4.51	46.73 ± 2.22	29.90 ± 2.09	55.74 ± 2.59	32.30 ± 3.02 ^a^
	80	46.46 ± 2.73	55.48 ± 0.98	54.55 ± 6.09	36.46 ± 2.51	61.46 ± 2.38	35.28 ± 2.36 ^a^
ZnSO_4_ ^2^		43.02 ± 3.64	46.17 ± 3.23 ^b^	44.30 ± 8.38	28.85 ± 2.16 ^b^	54.46 ± 3.26 ^b^	27.27 ± 8.34 ^b^
Zn–Gly ^2^		43.32 ± 3.72	50.02 ± 6.05 ^a^	50.64 ± 5.96	33.18 ± 4.11 ^a^	58.60 ± 3.84 ^a^	33.79 ± 2.96 ^a^
	0 ^3^	40.93 ± 3.15 ^b^	45.47 ± 3.91 ^b^	43.32 ± 6.54 ^b^	28.38 ± 1.95 ^b^	53.81 ± 2.75 ^b^	26.38 ± 7.07 ^b^
	80 ^3^	45.38 ± 2.45 ^a^	50.73 ± 4.92 ^a^	52.53 ± 5.84 ^a^	33.77 ± 3.37 ^a^	59.35 ± 3.09 ^a^	34.68 ± 3.52 ^a^
*p*-value	Maternal zinc	0.347	0.029	0.056	<0.001	0.016	0.013
	Offspring zinc	0.005	0.009	0.009	<0.001	0.002	0.004
	Maternal zinc × offspring zinc	0.873	0.078	0.591	0.104	0.756	0.031

Different letters (a–b) represent significant differences between the groups. ^1^ Each value represents the mean ± SE of 15 replicates (*n* = 15). ^2^ Each value represents the mean ± SE of 30 replicates (*n* = 30). ^3^ Each value represents the mean ± SE of 30 replicates (*n* = 30).

**Table 5 antioxidants-11-02456-t005:** Effects of maternal Zn in conjunction with their offspring dietary Zn supplementation on serum antioxidant indexes in offspring broilers (mean ± S.E.).

Maternal Zinc	Offspring Zinc (mg/kg)	T-AOC (U/mg Prot)	MDA (nmol/mg Prot)	T-SOD (U/mg Prot)	CuZn-SOD (U/mg Prot)
21d					
ZnSO_4_ ^1^	0	8.14 ± 0.57	7.91 ± 0.25 ^a^	516.6 ± 17.43	262.0 ± 8.87
	80	11.69 ± 0.75	5.46 ± 0.24 ^c^	571.8 ± 7.09	271.1 ± 11.41
Zn–Gly ^1^	0	8.47 ± 0.23	6.00 ± 0.43 ^b^	568.1 ± 32.66	265.7 ± 9.58
	80	13.26 ± 1.07	5.15 ± 0.35 ^c^	600.9 ± 30.53	296.2 ± 16.67
ZnSO_4_ ^2^		9.72 ± 1.97 ^b^	6.51 ± 1.33	537.3 ± 31.69 ^b^	267.8 ± 11.08 ^b^
Zn–Gly ^2^		10.27 ± 2.55 ^a^	5.61 ± 0.59	583.4 ± 34.92 ^a^	284.5 ± 20.78 ^a^
	0 ^3^	8.31 ± 0.45 ^b^	6.58 ± 0.99	548.3 ± 37.48 ^b^	264.1 ± 8.89 ^b^
	80 ^3^	12.36 ± 1.17 ^a^	5.27 ± 0.34	592.1 ± 28.82 ^a^	284.5 ± 19.05 ^a^
*p*-value	Maternal zinc	0.012	<0.001	0.004	0.016
	Offspring zinc	<0.001	<0.001	0.002	0.002
	Maternal zinc × offspring zinc	0.077	<0.001	0.374	0.064
60d					
ZnSO_4_ ^1^	0	10.12 ± 1.07 ^c^	7.03 ± 0.32 ^a^	427.0 ± 4.64 ^c^	316. 7 ± 8.38
	80	12.07 ± 0.74 ^b^	6.90 ± 0.61 ^a^	615.8 ± 13.91 ^a^	331.0 ± 7.9
Zn–Gly ^1^	0	10.82 ± 0.85 ^c^	6.99 ± 0.76 ^a^	569.9 ± 36.59 ^b^	322.0 ± 9.67
	80	13.84 ± 1.18 ^a^	5.80 ± 0.33 ^b^	630.7 ± 19.40 ^a^	345.2 ± 19.76
ZnSO_4_ ^2^		11.26 ± 1.31 ^b^	6.97 ± 0.46 ^a^	559.2 ± 91.91	323.9 ± 10.77
Zn–Gly ^2^		12.16 ± 1.85 ^a^	6.24 ± 0.78 ^b^	610.5 ± 38.63	337.5 ± 20.06
	0 ^3^	10.47 ± 0.98 ^b^	7.01 ± 0.50 ^a^	498.5 ± 81.69	319.0 ± 8.63 ^b^
	80 ^3^	12.72 ± 1.24 ^a^	6.29 ± 0.73 ^b^	622.7 ± 17.71	339.5 ± 17.07 ^a^
*p*-value	Maternal zinc	0.009	0.046	<0.001	0.0187
	Offspring zinc	<0.001	0.025	<0.001	0.019
	Maternal zinc × offspring zinc	0.217	0.061	<0.001	0.538

T-AOC, Total antioxidation capability. MDA, Malondialdehyde. T-SOD, Total superoxide dismutase. CuZn-SOD, Copper and zinc superoxide dismutase. Different letters (a–c) represent significant differences between the groups. Superscripts referring to 1, 2, and 3 have the same meaning as Table 4.

**Table 6 antioxidants-11-02456-t006:** Effects of maternal Zn in conjunction with their offspring dietary Zn supplementation on liver antioxidant indexes in offspring broilers (mean ± S.E.).

Maternal Zinc	Offspring Zinc (mg/kg)	T-AOC (U/mg Prot)	MDA (nmol/mg Prot)	T-SOD (U/mg Prot)	CuZn-SOD (U/mg Prot)	MT (ng/g)
21d						
ZnSO_4_ ^1^	0	0.97 ± 0.08	0.75 ± 0.01 ^a^	383.0 ± 15.76	260.0 ± 5.21	34.15 ± 2.36
	80	1.15 ± 0.10	0.56 ± 0.03 ^bc^	431.7 ± 13.42	277.0 ± 3.84	42.96 ± 2.32
Zn–Gly ^1^	0	1.05 ± 0.07	0.58 ± 0.03 ^b^	413.2 ± 15.54	265.3 ± 4.98	36.38 ± 1.81
	80	1.19 ± 0.09	0.52 ± 0.02 ^c^	464.9 ± 16.92	287.5 ± 8.10	46.44 ± 2.23
ZnSO_4_ ^2^		1.05 ± 0.13	0.68 ± 0.10	411.4 ± 28.61 ^b^	267.2 ± 10.06 ^b^	38.82 ± 5.07 ^b^
Zn–Gly ^2^		1.15 ± 0.11	0.55 ± 0.04	433. 9 ± 30.71 ^a^	276.4 ± 13.30 ^a^	41.02 ± 5.56 ^a^
	0 ^3^	1.00 ± 0.09 ^b^	0.67 ± 0.09	399.5 ± 21.65 ^b^	262.9 ± 5.53 ^b^	35.19 ± 2.35 ^b^
	80 ^3^	1.17 ± 0.09 ^a^	0.54 ± 0.03	443.8 ± 21.78 ^a^	283.6 ± 8.46 ^a^	44.35 ± 2.82 ^a^
*p*-value	Maternal zinc	0.128	<0.001	<0.001	0.020	0.002
	Offspring zinc	<0.001	<0.001	<0.001	<0.001	<0.001
	Maternal zinc × offspring zinc	0.601	<0.001	0.824	0.403	0.451
60d						
ZnSO_4_ ^1^	0	1.03 ± 0.10 ^c^	0.80 ± 0.05	442.0 ± 12.45 ^c^	303.0 ± 1.37 ^c^	33.07 ± 1.21
	80	1.21 ± 0.09 ^b^	0.69 ± 0.05	554.3 ± 37.62 ^ab^	335.6 ± 2.57 ^b^	37.51 ± 1.75
Zn–Gly ^1^	0	1.06 ± 0.11 ^c^	0.72 ± 0.07	530.6 ± 26.56 ^b^	303.5 ± 5.51 ^c^	34.62 ± 1.84
	80	1.59 ± 0.05 ^a^	0.65 ± 0.09	563.0 ± 8.89 ^a^	355.8 ± 6.40 ^a^	41.51 ± 1.58
ZnSO_4_ ^2^		1.14 ± 0.13	0.76 ± 0.08	498.1 ± 63.99	317.0 ± 17.54	35.92 ± 2.69 ^b^
Zn–Gly ^2^		1.37 ± 0.28	0.68 ± 0.08	546.8 ± 25.37	320. 9 ± 26.71	37.84 ± 3.93 ^a^
	0 ^3^	1.05 ± 0.10	0.77 ± 0.07 ^a^	478.0 ± 49.25	303.3 ± 4.19	34.02 ± 1.76 ^b^
	80 ^3^	1.41 ± 0.21	0.67 ± 0.07 ^b^	558.0 ± 28.51	345.7 ± 11.86	39.26 ± 2.62 ^a^
*p*-value	Maternal zinc	<0.001	0.162	<0.001	0.001	<0.001
	Offspring zinc	<0.001	0.036	<0.001	<0.001	<0.001
	Maternal zinc × offspring zinc	<0.001	0.552	<0.001	0.001	0.065

T-AOC, Total antioxidation capability. MDA, Malondialdehyde. T-SOD, Total superoxide dismutase. CuZn-SOD, Copper and zinc superoxide dismutase. MT, metallothionein. Different letters (a–c) represent significant differences between the groups. Superscripts referring to 1, 2, and 3 have the same meaning as Table 4.

**Table 7 antioxidants-11-02456-t007:** Effects of maternal Zn in conjunction with their offspring dietary Zn supplementation on muscle antioxidant indexes in offspring broilers (mean ± S.E.).

Maternal Zinc	Offspring Zinc (mg/kg)	T-AOC (U/mg Prot)	MDA (nmol/mg Prot)	T-SOD (U/mg Prot)	CuZn-SOD (U/mg Prot)
21d					
ZnSO_4_ ^1^	0	0.09 ± 0.008 ^b^	0.47 ± 0.02 ^a^	86.42 ± 2.31 ^c^	52.59 ± 4.21 ^b^
	80	0.11 ± 0.009 ^a^	0.31 ± 0.01 ^c^	103.1 ± 6.58 ^b^	62.98 ± 2.95 ^a^
Zn–Gly ^1^	0	0.11 ± 0.001 ^a^	0.35 ± 0.03 ^b^	87.68 ± 6.18 ^c^	52.77 ± 1.39 ^b^
	80	0.11 ± 0.01 ^a^	0.24 ± 0.017 ^d^	111.6 ± 8.44 ^a^	65.45 ± 3.74 ^a^
ZnSO_4_ ^2^		0.097 ± 0.01 ^b^	0.38 ± 0.09 ^a^	95.51 ± 9.98	56.92 ± 6.44
Zn–Gly ^2^		0.111 ± 0.007 ^a^	0.30 ± 0.06 ^b^	101.3 ± 14.28	59.11 ± 7.15
	0 ^3^	0.10 ± 0.011 ^b^	0.40 ± 0.07 ^a^	87.11 ± 4.66 ^b^	52.67 ± 3.11 ^b^
	80 ^3^	0.11 ± 0.009 ^a^	0.27 ± 0.03 ^b^	107.9 ± 8.61 ^a^	64.33 ± 3.48 ^a^
*p*-value	Maternal zinc	0.012	<0.001	0.085	0.345
	Offspring zinc	0.012	<0.001	<0.001	<0.001
	Maternal zinc × offspring zinc	0.135	0.031	0.194	0.412
60d					
ZnSO_4_ ^1^	0	0.055 ± 0.004 ^d^	0.37 ± 0.007 ^a^	83.77 ± 7.37	50.55 ± 4.58
	80	0.078 ± 0.005 ^b^	0.22 ± 0.036 ^c^	86.03 ± 3.94	51.60 ± 3.00
Zn–Gly ^1^	0	0.068 ± 0.001 ^c^	0.32 ± 0.009 ^b^	84.39 ± 2.79	50.37 ± 4.41
	80	0.086 ± 0.006 ^a^	0.20 ± 0.024 ^c^	86.05 ± 4.60	55.10 ± 2.75
ZnSO_4_ ^2^		0.067 ± 0.013 ^b^	0.31 ± 0.08 ^a^	85.18 ± 5.08	50.88 ± 3.78
Zn–Gly ^2^		0.076 ± 0.010 ^a^	0.26 ± 0.06 ^b^	86.30 ± 3.83	52.83 ± 4.30
	0 ^3^	0.06 ± 0.007 ^b^	0.35 ± 0.026 ^a^	84.12 ± 4.70	50.45 ± 4.28
	80 ^3^	0.08 ± 0.007 ^a^	0.22 ± 0.029 ^b^	87.00 ± 3.92	53.35 ± 3.29
*p*-value	Maternal zinc	0.002	0.014	0.552	0.283
	Offspring zinc	<0.001	<0.001	0.209	0.099
	Maternal zinc × offspring zinc	0.390	0.207	0.739	0.330

T-AOC, Total antioxidation capability. MDA, Malondialdehyde. T-SOD, Total superoxide dismutase. CuZn-SOD, Copper and zinc superoxide dismutase. Different letters (a–d) represent significant differences between the groups. Superscripts referring to 1, 2, and 3 have the same meaning as Table 4.

**Table 8 antioxidants-11-02456-t008:** Effects of maternal Zn in conjunction with their offspring dietary Zn supplementation on cytokine secretion in serum of offspring broilers (mean ± S.E.).

Maternal Zinc	Offspring Zinc (mg/kg)	IL-1 (μg/L)	IL-2 (μg/L)	TNF-α (ng/L)
21d				
ZnSO_4_ ^1^	0	0.11 ± 0.01	0.13 ± 0.01	37.68 ± 1.20
	80	0.10 ± 0.01	0.13 ± 0.02	36.63 ± 2.24
Zn–Gly ^1^	0	0.11 ± 0.01	0.13 ± 0.02	36.75 ± 2.20
	80	0.10 ± 0.01	0.14 ± 0.01	36.57 ± 0.75
ZnSO_4_ ^2^		0.11 ± 0.01	0.13 ± 0.01	37.15 ± 1.84
Zn–Gly ^2^		0.10 ± 0.01	0.14 ± 0.02	36.66 ± 1.61
	0 ^3^	0.11 ± 0.01	0.13 ± 0.02	37.22 ± 1.80
	80 ^3^	0.10 ± 0.01	0.14 ± 0.01	36.60 ± 1.63
*p*-value	Maternal zinc	0.829	0.279	0.374
	Offspring zinc	0.314	0.293	0.266
	Maternal zinc × offspring zinc	0.846	0.568	0.429
60d				
ZnSO_4_ ^1^	0	0.17 ± 0.03 ^a^	0.15 ± 0.02	48.62 ± 6.06 ^a^
	80	0.13 ± 0.02 ^b^	0.16 ± 0.03	40.30 ± 2.11 ^b^
Zn–Gly ^1^	0	0.14 ± 0.02 ^b^	0.15 ± 0.02	41.60 ± 3.23 ^b^
	80	0.13 ± 0.01 ^b^	0.17 ± 0.02	38.70 ± 1.85 ^b^
ZnSO_4_ ^2^		0.15 ± 0.03 ^a^	0.15 ± 0.02	44.46 ± 6.14 ^a^
Zn–Gly ^2^		0.13 ± 0.02 ^b^	0.16 ± 0.02	40.07 ± 2.93 ^b^
	0 ^3^	0.15 ± 0.03 ^a^	0.15 ± 0.02	45.29 ± 6.00 ^a^
	80 ^3^	0.13 ± 0.01 ^b^	0.16 ± 0.03	39.50 ± 2.10 ^b^
*p*-value	Maternal zinc	0.006	0.255	0.001
	Offspring zinc	0.001	0.063	<0.001
	Maternal zinc × offspring zinc	0.012	0.741	0.030

IL-1, interleukin-1. IL-2, interleukin-2. TNF-α, Tumor necrosis factor-α. Different letters (a–b) represent significant differences between the groups. Superscripts referring to 1, 2, and 3 have the same meaning as Table 4.

**Table 9 antioxidants-11-02456-t009:** Effects of maternal Zn in conjunction with their offspring dietary Zn supplementation on immune indexes in serum of offspring broilers (mean ± S.E.).

Maternal Zinc	Offspring Zinc (mg/kg)	IgG (μg/mL)	IgA (μg/mL)	IgM (μg/mL)
21d				
ZnSO_4_ ^1^	0	0.82 ± 0.08	0.45 ± 0.01 ^b^	6.68 ± 0.54 ^b^
	80	0.88 ± 0.11	0.47 ± 0.02 ^a^	7.39 ± 0.95 ^ab^
Zn–Gly ^1^	0	0.83 ± 0.10	0.46 ± 0.01 ^b^	6.86 ± 0.39 ^ab^
	80	0.89 ± 0.09	0.47 ± 0.02 ^a^	7.69 ± 1.14 ^a^
ZnSO_4_ ^2^		0.85 ± 0.10	0.46 ± 0.02	7.03 ± 0.83 ^b^
Zn–Gly ^2^		0.86 ± 0.09	0.46 ± 0.01	7.27 ± 0.93 ^a^
	0 ^3^	0.83 ± 0.09	0.45 ± 0.01 ^b^	6.77 ± 0.47
	80 ^3^	0.88 ± 0.10	0.47 ± 0.02 ^a^	7.54 ± 1.03
*p*-value	maternal zinc	0.830	0.234	0.357
	Offspring zinc	0.072	<0.001	0.005
	Maternal zinc × offspring zinc	0.989	0.508	0.817
60d				
ZnSO_4_ ^1^	0	0.92 ± 0.13 ^b^	0.46 ± 0.01	6.17 ± 0.48 ^c^
	80	1.01 ± 0.12 ^ab^	0.46 ± 0.01	7.42 ± 0.76 ^a^
Zn–Gly ^1^	0	0.96 ± 0.09 ^ab^	0.46 ± 0.02	6.83 ± 0.43 ^b^
	80	1.05 ± 0.13 ^a^	0.47 ± 0.01	7.71 ± 0.66 ^a^
ZnSO_4_ ^2^		0.97 ± 0.13	0.46 ± 0.01	6.80 ± 0.89 ^b^
Zn–Gly ^2^		1.01 ± 0.12	0.46 ± 0.02	7.29 ± 0.71 ^a^
	0 ^3^	0.94 ± 0.11 ^b^	0.46 ± 0.01	6.48 ± 0.56 ^b^
	80 ^3^	1.03 ± 0.12 ^a^	0.47 ± 0.02	7.56 ± 0.71 ^a^
*p*-value	maternal zinc	0.274	0.452	0.020
	Offspring zinc	0.032	0.084	<0.001
	Maternal zinc × offspring zinc	0.988	0.887	0.337

Immunoglobulin A. IgG, Immunoglobulin G. IgM, Immunoglobulin M. Different letters (a–c) represent significant differences between the groups. Superscripts referring to 1, 2, and 3 have the same meaning as Table 4.

**Table 10 antioxidants-11-02456-t010:** Effects of maternal Zn in conjunction with offspring dietary Zn supplementation on stress indexes in serum of offspring broilers (mean ± S.E.).

Maternal Zinc	Offspring Zinc (mg/kg)	21d			60d		
		CORT (μg/L)	HSP70 (μg/L)	CK (μg/L)	CORT (μg/L)	HSP70 (μg/L)	CK (μg/L)
ZnSO_4_ ^1^	0	419.5 ± 47.44	0.19 ± 0.02	3.16 ± 0.43	553.7 ± 62.68 ^a^	0.25 ± 0.02 ^b^	3.87 ± 0.52 ^a^
	80	383.3 ± 36.50	0.21 ± 0.01	2.99 ± 0.35	457.5 ± 38.99 ^b^	0.26 ± 0.03 ^b^	3.60 ± 0.28 ^ab^
Zn–Gly ^1^	0	405.2 ± 55.05	0.21 ± 0.02	3.08 ± 0.38	473.9 ± 54.37 ^b^	0.26 ± 0.03 ^b^	3.62 ± 0.35 ^ab^
	80	363.5 ± 28.53	0.21 ± 0.03	2.93 ± 0.26	448.5 ± 39.66 ^b^	0.31 ± 0.03 ^a^	3.43 ± 0.34 ^b^
ZnSO_4_ ^2^		401.4 ± 45.20	0.20 ± 0.02	3.07 ± 0.39	505.6 ± 70.82	0.26 ± 0.02	3.73 ± 0.43
Zn–Gly ^2^		384.4 ± 47.74	0.21 ± 0.02	3.00 ± 0.33	460.5 ± 47.64	0.28 ± 0.043	3.53 ± 0.35
	0 ^3^	412.4 ± 50.56 ^a^	0.20 ± 0.02	3.12 ± 0.40	515.9 ± 70.40	0.25 ± 0.03	3.75 ± 0.45
	80 ^3^	373.4 ± 33.46 ^b^	0.21 ± 0.02	2.96 ± 0.30	453.0 ± 38.56	0.28 ± 0.03	3.52 ± 0.32
*p*-value	Maternal zinc	0.219	0.162	0.536	0.009	0.009	0.101
	Offspring zinc	0.007	0.076	0.175	0.001	0.002	0.065
	Maternal zinc × offspring zinc	0.842	0.221	0.931	0.033	0.021	0.742

CORT, corticosterone. HSP70, heat shock protein 70. CK, creatine kinase. Different letters (a–b) represent significant differences between the groups. Superscripts referring to 1, 2, and 3 have the same meaning as Table 4.

## Data Availability

Data is contained within the article.

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
