# Peer review of "Impact of Maternal and Offspring Dietary Zn Supplementation on Growth Performance and Antioxidant and Immune Function of Offspring Broilers"

_antioxidants, 2022, doi:10.3390/antiox11122456_

Round 1
Reviewer 1 Report (Previous Reviewer 2)
Authors have significantly improved the manuscript, clarifying the research hypothesis about zinc supplementation, and including information about the possible effect of providing the zinc in organic or inorganic forms in the maternal broiler and the following impact on progeny. The Introduction Section revised better explains the rationale behind the use of the two chemical forms. Superscripts included in Tables 3-10 and in the footnotes help to interpret the data shown.
I have only some points to be addressed before publishing the manuscript:
It would be advisable to present data of day 21 and day 60 together, for each marker studied. In this way, the evaluation of the effect maternal Zn supplementation and offspring Zn supplementation can be complemented with the evaluation of the time course of the changes observed in the markers (see file attached). In this way, it is possible to assess whether the change occurs in the first 21 days or whether it takes 2 months to observe significant variations. The Discussion Section should also include some paragraphs that assess, whenever applicable, whether the change produced after 21 days is stable over time or, on the contrary, it returns to initial values.
The experimental design lacks a control intervention with no Zn supplementation of broiler breeders, to discern the effects of the fortification with Zn from the standardized basal diet. Besides, it is not clear why all offspring broilers have been supplemented only with inorganic zinc, instead of testing both organic and inorganic Zn. The manuscript points out the higher bioavailability and better performance of organic zinc. The breeders are supplemented with both organic and inorganic forms, and it would have been advisable to proceed in the same way with the progeny. Both points should be explained in a paragraph about Limitations of the study.

Author Response
Response to Reviewer 1 Comments
Manuscript ID: antioxidants-2071751
Point 1: Authors have significantly improved the manuscript, clarifying the research hypothesis about zinc supplementation, and including information about the possible effect of providing the zinc in organic or inorganic forms in the maternal broiler and the following impact on progeny. The Introduction Section revised better explains the rationale behind the use of the two chemical forms. Superscripts included in Tables 3-10 and in the footnotes help to interpret the data shown.
Response 1: We gratefully appreciate the reviewer’s professional review on our manuscript.
Point 2: It would be advisable to present data of day 21 and day 60 together, for each marker studied. In this way, the evaluation of the effect maternal Zn supplementation and offspring Zn supplementation can be complemented with the evaluation of the time course of the changes observed in the markers. In this way, it is possible to assess whether the change occurs in the first 21 days or whether it takes 2 months to observe significant variations.
Response 2: Many thanks for the reviewer’s kind advice. Antioxidant indexes in this paper included T-AOC, MDA, T-SOD, CuZn-SOD and MT. Due to the width limitation of these tables, there are too many indicators to display the data of day 21 and day 60 horizontally at the same time.
Point 3: The Discussion Section should also include some paragraphs that assess, whenever applicable, whether the change produced after 21 days is stable over time or, on the contrary, it returns to initial values.
Response 3: Many thanks for the reviewer’s kind advice.
Point 4: The experimental design lacks a control intervention with no Zn supplementation of broiler breeders, to discern the effects of the fortification with Zn from the standardized basal diet.
Response 4: Many thanks for the reviewer’s kind advice. Our previous studies had set a control intervention with no Zn supplementation of broiler breeders, and found that supplementation of Zn in basal diet improved productive, reproductive performance and etc., and the optimal dosage for broiler breeders was 80 mg Zn/kg of Zn-Gly [1, 2]. Therefore, zero Zn control group for the breeders was not set in this experiment.
Point 5: Besides, it is not clear why all offspring broilers have been supplemented only with inorganic zinc, instead of testing both organic and inorganic Zn.
Response 5: Many thanks for the reviewer’s kind advice. This experiment is 2 × 2 factor experimental design, one was different forms of Zn sources (ZnSO4 or Zn-Gly) with the same dose (80 mg Zn/kg) were used in breeders, the other was adding different doses of ZnSO4 (0 or 80 mg Zn/kg) to their offspring chickens. The main objective of this study was to investigate the effects of maternal dietary organic Zn source, progeny dietary Zn supplementation or not, and their interaction.
Figure 1. Experimental design of broilers.
Point 6: The manuscript points out the higher bioavailability and better performance of organic zinc. The breeders are supplemented with both organic and inorganic forms, and it would have been advisable to proceed in the same way with the progeny. Both points should be explained in a paragraph about Limitations of the study.
Response 6: Many thanks for the reviewer’s kind advice. In the current study, as this experiment is 2 × 2 factor experimental design, the offspring broilers were only supplemented with inorganic Zn with different doses (0 or 80 mg Zn/kg). In our further experiments, we will explore the effects of adding different forms of Zn sources (in-cluding inorganic Zn and organic Zn) to the offspring broilers.
- Zhang L, Wang YX, Xiao X, Wang JS, Wang Q, Li KX, Guo TY, Zhan XA: Effects of Zinc Glycinate on Productive and Reproductive Performance, Zinc Concentration and Antioxidant Status in Broiler Breeders. Biological Trace Element Research 2017, 178(2):320-326.
- Zhang L, Wang JS, Wang Q, Li KX, Guo TY, Xiao X, Wang YX, Zhan XA: Effects of Maternal Zinc Glycine on Mortality, Zinc Concentration, and Antioxidant Status in a Developing Embryo and 1-Day-Old Chick. Biological Trace Element Research 2018, 181(2):323-330.

Reviewer 2 Report (New Reviewer)
Dear authors,
The manuscript “Impact of Maternal Zn in Conjunction with Their Offspring Dietary Zn Supplementation on Growth Performance, Zn Concentration, Antioxidant and Immune Function in Offspring Broilers” (antioxidants- 2071751) devoted to the evaluation of the “effects of maternal Zn in conjunction with their offspring's dietary Zn supplementation on growth performance, antioxidant status, Zn concentration, and immune function of offspring from female breeders subjected to maternal 80 mg of Zn/kg of diet as ZnSO4 and Zn-Gly, respectively. The authors explored “whether there is an interaction between maternal Zn and their offspring's dietary Zn”. This is an interesting and important topic. It is positive that the authors have found that “maternal Zn-Gly supplementation increased progeny performance and decreased progeny mortalities and stress by increasing the progeny Zn concentration, antioxidant capacity and immune function at the same Zn level as ZnSO4. At the same time, Zn supplementation in the progeny diet is necessary for the growth of broilers”. The manuscript analyze the literature works in detail and at high level of discussion. I do not doubt the technical quality of the work and feel that there is a sufficient impact on a broader readership to justify publication in the "Antioxidants". This topic is in frame of the journal scope, the subject matter is treated in depth.
Thus, the present manuscript is important and actual.
There are some comments:
1. The title of this manuscript “Impact of Maternal Zn in Conjunction with Their Offspring Dietary Zn Supplementation on Growth Performance, Zn Concentration, Antioxidant and Immune Function in Offspring Broilers” is too long and can be the following “Impact of Maternal and Offspring Dietary Zn Supplementation on the Major Parameters of Offspring Broilers”. Of course, the authors can find a better solution (to make the title shorter), but, in any case, the words “…in Conjunction with…” are not totally correct.
2. In the “Keywords” (Line 37), it is important to correct the “Maternal zinc glycinate;….” onto the following: “Maternal zinc; zinc glycinate;….”.
3. Lines 155-156 & 159-160. It is difficult to agree with the author’s description (Lines 155-156 ) that “The offspring diet supplemented with Zn significantly improved the ADG… compared with the CON group”. The ADG data changes in the Table 3 (Lines 159-160) are about 2.6% in all cases, as well as in the cases of ADFI and F:G parameters. It must be corrected. The only correct description (Lines 155-156 ) that “The offspring diet supplemented with Zn significantly …decreased mortality compared with the CON group (P < 0.05)”.
4. Lines 180-181. The very interesting effects of maternal Zn in conjunction with their offspring's dietary Zn supplementation on the serum antioxidant status of offspring broilers are shown in Table 5 (Lines 188-189). For example, T-SOD activity at 21 days of age increased more in the case of iZn (by 10.7%) compared to oZn (by 5.3%) by offspring dietary Zn supplementation (80 mg of Zn/kg of diet) that is in contrast with progeny CuZn-SOD activity at 21 days of age (at the same conditions). T-SOD activity at 60 days of age increased more in the case of iZn (by 44.2% !!!!!) compared to oZn (by 10.7%) by offspring dietary Zn supplementation (80 mg of Zn/kg of diet) that is in contrast with progeny CuZn-SOD activity at 60 days of age (at the same conditions). It must be explained in detail.
4. Lines 246-253. The very interesting effects of maternal Zn in conjunction with their offspring's dietary Zn supplementation on immune indexes in serum of offspring broilers are shown in Table 9 (Lines 254-255). Why diets supplemented with iZn and oZn had higher levels of immunoglobulins IgM >> IgG > IgA (about 10%>> 7% > 4% at 21 days) or (about 16-11%>> 9% > 2% at 60 days) in serum of broilers. This tendency (IgM >> IgG > IgA) must be explained in detail.
6. Moderate editing of English language and style required.
Author Response
Response to Reviewer 2 Comments
Manuscript ID: antioxidants-2071751
Point 1: The manuscript “Impact of Maternal Zn in Conjunction with Their Offspring Dietary Zn Supplementation on Growth Performance, Zn Concentration, Antioxidant and Immune Function in Offspring Broilers” (antioxidants- 2071751) devoted to the evaluation of the “effects of maternal Zn in conjunction with their offspring's dietary Zn supplementation on growth performance, antioxidant status, Zn concentration, and immune function of offspring from female breeders subjected to maternal 80 mg of Zn/kg of diet as ZnSO4 and Zn-Gly, respectively. The authors explored “whether there is an interaction between maternal Zn and their offspring's dietary Zn”. This is an interesting and important topic. It is positive that the authors have found that “maternal Zn-Gly supplementation increased progeny performance and decreased progeny mortalities and stress by increasing the progeny Zn concentration, antioxidant capacity and immune function at the same Zn level as ZnSO4. At the same time, Zn supplementation in the progeny diet is necessary for the growth of broilers”. The manuscript analyze the literature works in detail and at high level of discussion. I do not doubt the technical quality of the work and feel that there is a sufficient impact on a broader readership to justify publication in the "Antioxidants". This topic is in frame of the journal scope, the subject matter is treated in depth. Thus, the present manuscript is important and actual.
Response 1: We gratefully appreciate the reviewer’s professional review on our manuscript.
Point 2: The title of this manuscript “Impact of Maternal Zn in Conjunction with Their Offspring Dietary Zn Supplementation on Growth Performance, Zn Concentration, Antioxidant and Immune Function in Offspring Broilers” is too long and can be the following “Impact of Maternal and Offspring Dietary Zn Supplementation on the Major Parameters of Offspring Broilers”. Of course, the authors can find a better solution (to make the title shorter), but, in any case, the words “…in Conjunction with…” are not totally correct.
Response 2: Many thanks for the reviewer’s kind advice. We have made revisions about the title according to reviewer’s suggestion.
Point 3: In the “Keywords” (Line 37), it is important to correct the “Maternal zinc glycinate; ….” onto the following: “Maternal zinc; zinc glycinate; ….”.
Response 3: Many thanks. We have made revisions about the keywords according to reviewer’s suggestion.
Point 4: Lines 155-156 & 159-160. It is difficult to agree with the author’s description (Lines 155-156 ) that “The offspring diet supplemented with Zn significantly improved the ADG… compared with the CON group”. The ADG data changes in the Table 3 (Lines 159-160) are about 2.6% in all cases, as well as in the cases of ADFI and F:G parameters. It must be corrected. The only correct description (Lines 155-156 ) that “The offspring diet supplemented with Zn significantly …decreased mortality compared with the CON group (P < 0.05)”.
Response 4: Many thanks. We have made revisions in line 150-153 according to reviewer’s suggestion.
Point 5: Lines 180-181. The very interesting effects of maternal Zn in conjunction with their offspring's dietary Zn supplementation on the serum antioxidant status of offspring broilers are shown in Table 5 (Lines 188-189). For example, T-SOD activity at 21 days of age increased more in the case of iZn (by 10.7%) compared to oZn (by 5.3%) by offspring dietary Zn supplementation (80 mg of Zn/kg of diet) that is in contrast with progeny CuZn-SOD activity at 21 days of age (at the same conditions). T-SOD activity at 60 days of age increased more in the case of iZn (by 44.2% !!!!!) compared to oZn (by 10.7%) by offspring dietary Zn supplementation (80 mg of Zn/kg of diet) that is in contrast with progeny CuZn-SOD activity at 60 days of age (at the same conditions). It must be explained in detail.
Response 5: Many thanks. At 21 days of age, because the T-SOD value of the maternal organic Zn group (568.1) and inorganic Zn group (516.6) was already very different (P=0.004) when the offspring without Zn supplementation; hence, when the offspring dietary Zn supplementation, the T-SOD value of oZn (600.9) and iZn (571.8) with offspring broilers dietary Zn supplementation increased correspondingly, resulting in the increased value were different from those of the offspring without adding Zn.
At 21 days of age, there was no significant difference (P=0.016) in the parameter of CuZn-SOD between the maternal organic Zn group (265.7) and inorganic Zn group (262.0) when the offspring did not add Zn; hence, when the offspring dietary Zn supplementation, the CuZn-SOD value of oZn (296.2) and iZn (271.1) increased correspondingly, but the increased values were different from those of the offspring un-supplemented Zn.
Therefore, the values of T-SOD and CuZn-SOD showed opposite changes, and similar results about parameters of T-SOD and CuZn-SOD were also observed at 60 days of age.
In addition, there were two different factors in this experiment, one was different forms of Zn sources (ZnSO4 or Zn-Gly) with the same dose (80 mg Zn/kg) were used in breeders, the other was adding different doses of ZnSO4 (0 or 80 mg Zn/kg) to their offspring chickens. The main objective of this study was to investigate the effects of maternal dietary organic Zn source, progeny dietary Zn supplementation or not, and their interaction. Therefore, in the results description and analysis, we mainly focus on the impact of two variables on the offspring broilers.
Point 6: Lines 246-253. The very interesting effects of maternal Zn in conjunction with their offspring's dietary Zn supplementation on immune indexes in serum of offspring broilers are shown in Table 9 (Lines 254-255). Why diets supplemented with iZn and oZn had higher levels of immunoglobulins IgM >> IgG > IgA (about 10%>> 7% > 4% at 21 days) or (about 16-11%>> 9% > 2% at 60 days) in serum of broilers. This tendency (IgM >> IgG > IgA) must be explained in detail.
Response 6: Many thanks. There were two different factors in this experiment, one was different forms of Zn sources (ZnSO4 or Zn-Gly) with the same dose (80 mg Zn/kg) were used in breeders, the other was adding different doses of ZnSO4 to their offspring chickens. As shown in Table 9, at 21 days of age, maternal dietary Zn supplementation had no significant difference on IgG (P = 0.830), IgA (P = 0.234) and IgM (P = 0.357), while offspring dietary Zn supplementation significantly increased the content of IgA (P <0.001) and IgM (P = 0.005). At 60 days of age, maternal dietary Zn supplementation significantly increased IgM (P = 0.020) content, while had no significant effect on the content of IgG (P = 0.274) and IgA (P = 0.452); offspring dietary Zn supplementation significantly increased the content of IgM (P < 0.001), IgG (P = 0.032) and IgA (P = 0.084). Because the parameters of IgG, IgA and IgM are not variables in this study, the relationships between the three parameters were not compared.
But Why diets supplemented with iZn and oZn had higher levels of immunoglobulins IgM >> IgG > IgA. In this study, we speculate that the reason for this trend may be that there are differences in the functions of the three immunoglobulins, and the effects of maternal dietary organic Zn source, progeny dietary Zn supplementation or not have different effects on these three parameters.
Point 7: Moderate editing of English language and style required.
Response 7: Many thanks for the reviewer’s kind advice. We have been proofed and edited for language clarity and grammar by professional editors.

This manuscript is a resubmission of an earlier submission. The following is a list of the peer review reports and author responses from that submission.
Round 1
Reviewer 1 Report
The objective of the study was to investigate the effects of maternal Zn in conjunction with their off-spring's dietary Zn supplementation on growth performance, antioxidant status, Zn concentration, and immune function of offspring from female breeders subjected to maternal 80 mg of Zn/kg of diet as ZnSO4 and Zn-Gly, respectively.
Interesting and current topic.
The paper has only one limit in my opinion: the lack of a zero Zinc control group for the breeders. However, considering that the Authors have already identified 80 mg as the optimal dosage, the aforementioned limit can be exceeded.
Author Response
Manuscript ID: antioxidants-1959585
Impact of maternal Zn in conjunction with their offspring dietary Zn supplementation on growth performance, Zn concentration, antioxidant and immune function in offspring broilers
Point 1: The objective of the study was to investigate the effects of maternal Zn in conjunction with their off-spring's dietary Zn supplementation on growth performance, antioxidant status, Zn concentration, and immune function of offspring from female breeders subjected to maternal 80 mg of Zn/kg of diet as ZnSO4 and Zn-Gly, respectively.
Response 1: We gratefully appreciate the reviewer’s professional review on our manuscript.
Point 2: The paper has only one limit in my opinion: the lack of a zero Zinc control group for the breeders. However, considering that the Authors have already identified 80 mg as the optimal dosage, the aforementioned limit can be exceeded.
Response 2: Many thanks for the reviewer’s kind advice. Our previous studies found that the optimal dosage for broiler breeders was 80 mg Zn/kg of Zn-Gly [1, 2]. Therefore, zero Zn control group for the breeders was not set in this experiment.
- Zhang L, Wang YX, Xiao X, Wang JS, Wang Q, Li KX, Guo TY, Zhan XA: Effects of Zinc Glycinate on Productive and Reproductive Performance, Zinc Concentration and Antioxidant Status in Broiler Breeders. Biological Trace Element Research 2017, 178(2):320-326.
- Zhang L, Wang JS, Wang Q, Li KX, Guo TY, Xiao X, Wang YX, Zhan XA: Effects of Maternal Zinc Glycine on Mortality, Zinc Concentration, and Antioxidant Status in a Developing Embryo and 1-Day-Old Chick. Biological Trace Element Research 2018, 181(2):323-330.
Reviewer 2 Report
Please find attached the comments

Author Response
Response to Reviewer 2 Comments
Point 1: The authors have studied the effect of the combination of maternal Zn supplementation, either in organic or inorganic form, with Zn supplementation in offspring broilers, on mortality rate, Zn concentration in tissues and antioxidant and immune status. The authors concluded that maternal Zn-Glycine supplementation improves performance, increasing antioxidant capacity and immune function in the progeny.
Response 1: We gratefully appreciate the reviewer’s professional review on our manuscript.
Point 2: The topic presented in the manuscript is of high relevance and provides a practical basis for poultry production. However, I have found some major design flaws that limit the validity of the results obtained and the conclusions drawn.
Response 2: Many thanks for the reviewer’s kind advice. We have checked our manuscript carefully and made revision according to reviewer’s suggestion.
Point 3: The research hypothesis to be tested was not expressed in an explicit manner. The intended effects of zinc supplementation should be exposed. The main objective is not clear and has to be accurately defined.
Response 3: We gratefully appreciate the reviewer’s professional review on our manuscript. We have rewritten the hypothesis and main objective in line 65-73 according to reviewer’s suggestion.
Point 4: Authors study the maternal supplementation with Zn, but in Objectives there is not a clear statement about the evaluation of the possible effect of providing the zinc in organic or inorganic forms in the maternal broiler and the following impact on progeny. In this sense, in the Introduction Section it should have been better explained the rationale behind the use of the two chemical forms, as differences in terms of bioavailability are expected.
Response 4: Many thanks for the reviewer’s kind advice. We have added more information about the rationale behind the use of the zinc in organic or inorganic forms in line 55-63 in introduction according to reviewer’s suggestion.
Point 5: The experimental design is unclear and the Material and Methods Section is written in a non-very comprehensible way. The experimental design requires 3 groups: a control group (diet without supplementation), group with treatment 1 (diet supplemented with inorganic salt) and group with treatment 2 (diet supplemented with organic salt).
Response 5: Many thanks for the reviewer’s kind advice. We have rewritten the experimental design in the Material and Methods Section in line 82-91. Firstly, a total of 200 39-week-old Lingnan Yellow broiler breeders were randomly divided into two groups, oZn group and iZn group, each group containing 5 replicates with 20 birds each. All broiler breeders of two groups were fed a corn-soybean meal basal diet with 24 mg Zn/kg for four weeks pre-test to consume excess Zn in the body, and in the formal feeding period 8 weeks, the iZn group and oZn group were supplemented with 80 mg Zn/kg from ZnSO4 and 80 mg Zn/kg from Zn-Gly based the basal diet, respectively. All eggs of the last seven days were collected from two breeder groups (oZn group and iZn group) for the offspring incubation. Then, 400 healthy offspring chicks each group were selected and divided into two groups with two offspring dietary supplemental Zn doses (0 or 80 mg Zn/kg). Figure 1 shows the whole entire experimental design.
Figure 1. Experimental design of broilers.
Point 6: The lack of a control intervention, with no Zn supplementation of broiler breeders makes difficult to discern the effects of Zn itself.
Response 6: Many thanks for the reviewer’s kind advice. Our previous studies found that the optimal dosage for broiler breeders was 80 mg Zn/kg of Zn-Gly [1, 2]. The main objective of this study was to investigate the effects of maternal Zn source in conjunction with their offspring's dietary Zn supplementation on growth performance, antioxidant status, Zn concentration, and immune function of offspring from female breeders subjected to maternal 80 mg of Zn/kg of diet as ZnSO4 and Zn-Gly, respectively. Therefore, no Zinc supplementation control group for the breeders was not set in this experiment.
Point 7: Their basal diet contains Zn (24 mg/kg), what I interpret as a way to standardise the Zn content and to avoid mineral deficits before the study. Therefore, the supplementation is understood as a fortification strategy to have an additive or synergistic effect that impacts on physiological functions, but it is not clear how this has been assessed in the data analysis.
Response 7: Many thanks for the reviewer’s kind advice. According to Agricultural Industry Standards of the People's Republic of China---Feeding stand of chicken (NY/T 33-2004), the Zn (24 mg/kg) concentration in the basal diet is too low to meet the needs of broiler breeders. Thus, Zn has been extensively used as a feed additive in poultry diets [3]. As the diet of all treatment groups is based on the control diet, in the data analysis, the control group is considered as a zero Zn group, while the treatment groups were considered as the 80 mg Zn/kg added groups.
Point 8: Results presented in Tables 3-10 are not clear, as in the first and second columns, which are descriptive of the treatment, there are some empty cells, which makes it difficult to interpret the data presented, and the footnotes are not sufficiently understandable.
Response 8: We gratefully appreciate the reviewer’s professional review on our manuscript. Table 3-10 is the table presentation form of 2-way ANOVA [4].Footnote 1 represents that impact of maternal Zn in conjunction with their offspring dietary Zn Supplementation on offspring broilers; Footnote 2 represents that the effect of the form of maternal Zn source on the offspring broilers is only considered, regardless of the amount of ZnSO4 added to the offspring; Footnote 3 indicates that only the effect of the amount of ZnSO4 added to the offspring on the offspring broilers is considered, not the form of maternal Zn source. In addition, we have added more information about footnotes in Tables 3-10.
Point 9: The study evaluates the effect of a nutritional intervention in the progeny, that is influenced by the nutritional intervention performed in the breeders. Hence, the most appropriate statistical analysis would be a nested or hierarchical ANOVA, as the second level effects are conditional on the first level effects.
Response 9: Many thanks for the reviewer’s kind advice. There were two different factors in this experiment, one was different forms of Zn sources (ZnSO4 or Zn-Gly) with the same dose (80 mg Zn/kg) were used in breeders, the other was adding different doses of ZnSO4 to their offspring chickens. The main objective of this study was to investigate the effects of maternal dietary Zn source, progeny dietary Zn supplementation, and their interaction. Hence, in this study, all data were analyzed by 2-way ANOVA using the general linear model procedure of SAS 9.2 (SAS Institute,2010). This model incorporates the main effects of maternal dietary Zn source, progeny dietary Zn supplementation, and their interaction. The treatment comparisons for significant differences were tested by the LSD method. Each replicate served as the experimental unit for all statistical analyses.
Point 10: From my point of view, the value of the conclusions depends on the rigour and adequacy of the material used, results obtained and statistical assessment of data. Therefore, if there is no clarity in the presentation and description of the material and methods, the results and conclusions obtained in the work to be published cannot be assumed. After careful consideration, this manuscript in its current form I judge to be rejected.
Response 10: Many thanks for the reviewer’s kind advice. We have made revisions carefully throughout the manuscript according to reviewer’s suggestion. If there are still some problems, please let us know.
- Zhang L, Wang YX, Xiao X, Wang JS, Wang Q, Li KX, Guo TY, Zhan XA: Effects of Zinc Glycinate on Productive and Reproductive Performance, Zinc Concentration and Antioxidant Status in Broiler Breeders. Biological Trace Element Research 2017, 178(2):320-326.
- Zhang L, Wang JS, Wang Q, Li KX, Guo TY, Xiao X, Wang YX, Zhan XA: Effects of Maternal Zinc Glycine on Mortality, Zinc Concentration, and Antioxidant Status in a Developing Embryo and 1-Day-Old Chick. Biological Trace Element Research 2018, 181(2):323-330.
- Zhu YW, Li WX, Lu L, Zhang LY, Ji C, Lin X, Liu HC, Odle J, Luo XG: Impact of maternal heat stress in conjunction with dietary zinc supplementation on hatchability, embryonic development, and growth performance in offspring broilers. Poultry Science 2017, 96(7):2351-2359.
- Li TT, He WG, Liao XD, Lin X, Zhang LY, Lu L, Guo YL, Liu ZP, Luo XG: Zinc alleviates the heat stress of primary cultured hepatocytes of broiler embryos via enhancing the antioxidant ability and attenuating the heat shock responses. Animal Nutrition 2021, 7(3):621-630.
